# Spatiotemporal Analysis for COVID-19 Delta Variant Using GIS-Based Air Parameter and Spatial Modeling

**DOI:** 10.3390/ijerph19031614

**Published:** 2022-01-30

**Authors:** Mokhamad Nur Cahyadi, Hepi Hapsari Handayani, IDAA Warmadewanthi, Catur Aries Rokhmana, Soni Sunarso Sulistiawan, Christrijogo Sumartono Waloedjo, Agus Budi Raharjo, Mohamad Atok, Shilvy Choiriyatun Navisa, Mega Wulansari, Shuanggen Jin

**Affiliations:** 1Geomatics Engineering Department, Institut Teknologi Sepuluh Nopember, Surabaya 60111, Indonesia; hapsari@geodesy.its.ac.id (H.H.H.); shilvynavisa.18033@mhs.its.ac.id (S.C.N.); mega.18033@mhs.its.ac.id (M.W.); 2Research Center Science Technology of Marine and Earth, Institut Teknologi Sepuluh Nopember, Surabaya 60111, Indonesia; 3Department of Environmental Engineering, Institut Teknologi Sepuluh Nopember, Surabaya 60111, Indonesia; warma@its.ac.id; 4Faculty of Engineering, Universitas Gadjah Mada, Yogyakarta 5581, Indonesia; caris@ugm.ac.id; 5Department of Anesthesiology and Reanimation, Faculty Of Medicine, Universitas Airlangga-Dr. Soetomo Hospital, Surabaya 60132, Indonesia; soni.sunarso.s@gmail.com; 6Master Program in Disaster Management, Postgraduate School, Universitas Airlangga-Dr. Soetomo Hospital, Surabaya 60132, Indonesia; christrijogo@fk.unair.ac.id; 7Department of Informatics Engineering, Institut Teknologi Sepuluh Nopember, Surabaya 60111, Indonesia; agus.budi@its.ac.id; 8Department of Electrical Engineering, Institut Teknologi Sepuluh Nopember, Surabaya 60111, Indonesia; endroyono@ee.its.ac.id; 9Department of Statistics, Institut Teknologi Sepuluh Nopember, Surabaya 60111, Indonesia; moh_atok@statistika.its.ac.id; 10Shanghai Astronomical Observatory, Chinese Academy of Sciences, Shanghai 200030, China; sgjin@shao.ac.cn

**Keywords:** COVID-19, lockdown, spatial pattern, air pollutions

## Abstract

The coronavirus disease of 2019 (COVID-19) pandemic is currently a global challenge, with 210 countries, including Indonesia, seeking to minimize its spread. Therefore, this study aims to determine the spatiotemporal spread pattern of this virus in Surabaya using various data on confirmed cases from 28 April to 26 October 2021. It also aims to determine the relationship between pollutant parameters, such as carbon monoxide (CO), nitrogen dioxide (NO_2_), sulfur dioxide (SO_2_), and ozone (O_3_), as well as the government’s high social restrictions policy in Java-Bali. Several methods, such as the weighted mean center, directional distribution, Getis–Ord Gi*, Moran’s I, and geographically weighted regression, were used to identify the spatial spread pattern of the virus. The weighted mean center indicated that the epicenter location of the outbreak moved randomly. The directional distribution demonstrated a decrease of 21 km^2^ at the end of the study phase, which proved that its spread has significantly reduced in Surabaya. Meanwhile, the Getis–Ord Gi* results demonstrated that the eastern and southern parts of the study region were highly infected. Moran’s I demonstrate that COVID-19 cases clustered during the spike. The geographically weighted regression model indicated a number of influence zones in the northeast, northwest, and a few in the southwest parts at the peak of R^2^ 0.55. The relationship between COVID-19 cases and air pollution parameters proved that people living at the outbreak’s center have low pollution levels due to lockdown. Furthermore, the lockdown policy reduced CO, NO_2_, SO_2_, and O_3_. In addition, increase in air pollutants; namely, NO_2_, CO, SO_2_ and O_3_, was recorded after 7 weeks of lockdown implementation (started from 18 August).

## 1. Introduction

According to the World Health Organization (WHO), the world has been battling the spread of the coronavirus disease of 2019 or the COVID-19 pandemic since March 2020, with over 118,319 positive cases and 4292 deaths globally [1]. This virus, which causes severe respiratory problems, has a high human-to-human transmission rate and requires technology capable of analyzing and determining its spread pattern [2]. COVID-19 is caused by a severe acute respiratory syndrome coronavirus 2 (SARS-CoV-2) [2]. There are seven genera of CoV, and SARS-CoV-2 is the seventh member of the human-infecting CoV family. SARS-CoV and MERS-CoV are responsible for pneumonia, while HCoV-229E, HCoV-NL63, HCoVOC43, and HCoV-HKU1 cause relatively mild and self-limiting respiratory symptoms (common cold) [2,3]. Several efforts have been made to control the spread of this virus at the local, national, and global scale. An example is the implementation of the lockdown policies, which vary across countries and cities worldwide. However, the lockdown significantly halted all forms of transport (flights, trains, and automobiles), factories, shops, markets, and other economic and social activities.

Surabaya, the capital of East Java, has the second largest population in Indonesia with 2,874,314 people [4] and it is the second largest city after Jakarta. It is also included as the largest metropolitan area in eastern Indonesia, which known as Gerbangkertosusila (Gresik–Bangkalan–Mojokerto–Surabaya–Sidoarjo-Lamongan) [5]. Surabaya is also the economic, commercial, industrial, and educational hub of East Java and Indonesia’s eastern region [5]. The National COVID-19 Task Force (Satuan Tugas (Satgas)) revealed on 20 June that East Java had the largest number of confirmed COVID-19 deaths, with a total of 12,074 deaths. According to the Satgas of East Java, the city of Surabaya had the highest number of deaths in the province, with a total of 1382 deaths [6].

By 9 April 2020, the pandemic had spread to all provinces in Indonesia, and on 24 June 2020, over 500 cases were confirmed in almost every region [7]. According to the information posted on Lawan’s website, [8] the Surabaya city area recorded 64,547 confirmed cases of COVID-19 on 26 August 2021.

Presently, the Indonesian government has made a series of policies to prevent the spread and transmission of the coronavirus from one community to another. These policies include global scale social restrictions (PSBB), micro PPKM, implementation of Java-Bali community activity restrictions (PPKM), and vaccination. Subsequently, the Indonesian Ministry of Health also issued guidelines in the subsequent development of the pandemic, which provided recommendations based on data adjustments to economic and social activities in the community through the contact-tracing mechanism. The delta variant in Indonesia was first detected in April 2021, and it is more transmissible with a greater possibility of causing severe symptoms [9]. WHO declared this variant as the fastest and most sensitive strain ever, with significant infectious strain, and capable of “picking up” the most susceptible individuals [6]. This is particularly troubling to those who have not been fully or completely vaccinated, as well as those with weak immune response systems.

A regional spatiotemporal understanding of COVID-19 is critical to provide insight into how the pandemic occurred and its continued growth and decline [10,11,12]. Previous studies by Purwanto et al. [10] demonstrated that it is important to determine the spatial and temporal resolution of the STC model because it affects the detailed information on the endemic/epidemic, and it is important for the hotspot analysis results. Al-Kindi et al. [11] also used the spatiotemporal assessment to determine the virus spread in Oman for 2 months. The study demonstrated that the directional pattern of COVID-19 cases has moved from the northeast to the northwest and southwest, increasing the total affected area over time. The results also demonstrated that the virus distribution is higher in the most densely populated areas. COVID-19 spatiotemporal characteristics and trends need to be considered when scheduling the reopening of activities within a state or location [12].

The virus spread can be caused by several factors, asides from the spatial variable. A study by Hassan et al. [13] demonstrated the spatial relationships between COVID-19 cases and several variables, such as air pollution, geo-meteorological, and social parameters. The study found a significant robust relationship between those variables in cities with significant positive cases. In addition, the correlation finding suggests that long-term bad air quality may aggravate the clinical symptoms of the disease [14].

Despite the severe impacts of lockdown on people’s social life, global mobility, and economy, it temporarily improved environmental conditions. For instance, based on satellite images, many media reported a decrease in air pollution globally. This was also found in the study by He et al. and Cadotte [15,16] conducted in some countries. Broomandi et al. [17] outlined a relationship between lockdowns and air pollution, where a decrease in road traffic and economic activities reduced the level of carbon monoxide (CO), nitrogen dioxide (NO_2_), sulfur dioxide (SO_2_), and particulate matter (PM10) in Tehran, irrespective of the unfavorable weather conditions. In contrast, the Ozone (O_3_) and PM 2.5 concentrations increased.

However, no research has been conducted on the concurrent correlation between spatial, temporal, and air pollution. Therefore, this study aims to (1) assess the spatiotemporal pattern of COVID-19 using updated daily data recorded for 26 weeks, and (2) investigate various factors, such as confirmed cases, recovered patients, suspected cases, and air pollution (CO, SO_2_, O_3_, and NO parameters) by conducting geographically weighted regression (GWR) and ordinary least squares (OLS) regression. It aims to (3) analyze the relationship between air pollution concentrations and COVID-19 cases for 26 weeks, estimated by Sentinel-5P imagery. This research focused on the Surabaya city as one of the areas with a high death rate during the delta variant period, according to a WHO report on 23 June 2021 [6]. Therefore, the result is expected to help the Surabaya government create more appropriate policies and strategies to reduce the virus spread.

## 2. Materials and Methods

### 2.1. Study Area

Surabaya is geographically located at 7°9′–7°21′ S and 112°36′–112°57′ E, with an area of 52,087 Ha. Its land and sea areas are 63.45% (33,048 Ha) and 36.55% (19,039 Ha) wide and managed by the government. The area is divided into 154 villages, as shown in Appendix A. In 2020, the total population recorded was 3,148,939 people, with Wonokusumo and Romokalisari villages constituting the largest and least, respectively [18]. Appendix A shows the population and density of each issue in 2020.

### 2.2. Dataset

Data were collected from the official website of Satuan Tugas on COVID-19 cases in Surabaya from 28 April to 26 October 2021. This duration was selected because of the significant increase in confirmed cases at that timeframe [8]. These data were spatially linked as administrative boundaries, as shown in Table 1. All spatial data are integrated into a geodatabase file using ArcGIS (ESRI, Redlands, California). Figure 1 shows a graph of the additional COVID-19 confirmed cases for each week from 28 April to 26 October 2021 (the first week starts on 28 April and the 26th week ends on 26 October). The spatial distribution of positive cases per 100,000 populations in each village is shown in Figure 2. This also indicates the different hotspots during a pandemic as a function of the total population from each village. Air pollution quality was also analyzed, specifically in COVID-19 hotspots, using the Sentinel-5P satellite image data to obtain an estimate of tropospheric concentrations of NO_2_, SO_2_, CO, and O_3_.

### 2.3. Spatial Analytic Method

The geography information system has several tools used to determine each epidemic risk’s various spatial statistics, such as distribution, hotspot, orientation, and the trajectory of spread, etc. Its techniques were used to investigate the spatial variation of the virus, visualize related information, and track pandemic hotspots spatially during the study period. Besides, it is necessary to understand the spatial variability of the incidence in relation to different environmental, sociographic, and demographic variables. It is also for spatio-temporal prediction of the speed and magnitude of regional transmission in the near future.

#### 2.3.1. Calculating Geographic Distribution

Calculating geographic distribution (CGD) is a method used to measure the disease distribution and identify certain characteristics, such as center, direction, and orientation [11]. Epidemiologist experts use it to track and compare the changes in different places over time [19]. This study used ArcGIS 1.7 in accordance with the CGD method to analyze the spatial distribution of COVID-19 in Surabaya. CGD is used by calculating and tracking the distribution point weekly from 30 June to 24 August 2021. The first step was conducted to identify the mean center of the additional confirmed cases per week. It was performed based on the extra information related to daily cases. The second step was performed to calculate the weighted variation in the distance between each location. The directional distribution or DD (standard deviational ellipse) tool was used to track the direction and orientation of the mean center (MC) during the study period. This tool is mainly determined by average location, dispersion (concentration), and orientation [20]. Directional distribution (standard deviational ellipse) calculates two standard deviations, one along a transformed axis of maximum concentration (y) and the other along an axis that is orthogonal to (x) [21]. This ellipse provides the changing trend of COVID-19 over time.

#### 2.3.2. Spatially Integrated Statistics

Moran’s I and the General G test were the geospatial statistical methods used to analyze the spatial distribution pattern of COVID-19 in this study. Both methods were used to measure the autocorrelation of spatial data and determine the spatial clustering of COVID-19 [11]. Spatial data are simply described as highly correlated, assuming the spatial variables have values close to each other and are conversely defined as independent or random data when no pattern can be identified to explain its arrangement [22].

In GIS, Moran’s I autocorrelation is used to obtain positive and negative values indicating a clustering tendency and random distribution pattern [23]. However, other statistical attributes are the z-score, which quantifies the degree of deviation, such as dispersion or clustering around Moran’s I scores and p-value. In this context, significant autocorrelation demonstrates that the value of a variable at a given location depends on its worth at a neighboring site and vice versa [23]. Typically, the global Moran’s I value ranges from −1 to 1 and is used to denote a random pattern when close to −1 and clustering at 1 [24]. According to Prasannakumar et al. [25], Moran’s I is calculated using the following equation:(1)$$I=\frac{N\sum_{i}\sum_{j}W_{ij}\left(x_{i}-\bar{x}\right)\left(x_{j}-\bar{x}\right)}{(\sum_{i}\sum_{j}W_{ij})\sum_{j}\left(x_{i}-\bar{x}\right){\left(x_{j}-\bar{x}\right)}^{2}}$$
where N denotes the incidence of COVID-19, $x_{i}$ and $x_{j}$ are the variable values in different locations, $\bar{x}$ is the average of the variable, and $W_{ij}$ is the weight applied to the comparison between locations i and j. This distance-based weight matrix is in accordance with the inverse distance between locations I and j (1/dij).

Similarly, the General G test is another indicator of spatial autocorrelation used to identify hotspots and local spatial clustering of COVID-19 [11]. The Getis–Ord General G high/low clustering statistic was used to denote the occurrence of clustering and whether values are strongly above or below the average. Large positive General G values close to 1 imply that clustered values are higher than average, whereas large negative General G values close to −1 suggest that they are lower than normal [26]. The Z-score is derived as the difference between the actual and expected values for General G, divided by the standard deviation of the expected randomly distributed values [27]. General G is positive, assuming there are more high values than low, and negative, supposing there are more low values than high and when both tend to cluster. According to Getis and Aldstadt [28], G is calculated using the following equation:(2)$$G=\frac{\sum_{i=1}^{n}\sum_{j=1}^{n}w_{ij}x_{i}x_{j}}{\sum_{i=1}^{n}\sum_{j=1}^{n}x_{i}x_{j}}\;,\;\forall_{j}\neq i$$
where $x_{i}$ and $x_{j}$ are the attribute values for locations i and j, $w_{ij}$ denotes the weighted spatial distance between locations i and j, N is the number of locations, Ji depicts it, and i and j do not reflect the same feature. The G test usually provides 4 values, namely the observed and expected General Gs, the Z score, and the *p*-value [28]. The first step in analyzing the General G autocorrelation is to evaluate the p-value of these statistics. Moreover, when small and large, the cases experienced spatial grouping, and random distribution, respectively. The second step involves discerning the sign of the Z score. The positive sign indicates that the higher values of COVID-19 cases tend to cluster in the study domain when the observed General G index is greater than expected. In contrast, the negative sign of the Z score implies that lesser values tend to be clustered when the General G index is less than expected [29].

A hotspot analysis (Getis–Ord Gi) is commonly used to determine hotspots in GIS by providing 2 statistical values, namely Z score and *p*-value. According to Huang et al. [22] both statistical values indicate the possibility of the maximum and a minimum number of COVID-19 cases to have spatial dependence or clustering. Based on the z score, it is evident that high or low incidence locations tend to be spatially clustered [22]. Furthermore, every location considered a significant COVID-19 hotspot demonstrates a high incidence [29]. The number of cases in a specific point, including its neighbors, is compared proportionally to those at all points in the location. A statically significant positive Z score indicates high-intensity clustering values, determined when the local count is different from the expected number. According to Huling et al. [30], a higher Z score indicates more clustering of COVID-19 incidence (hotspots). On the contrary, a negative and statistically significant smaller Z score implies more clusters of cold spots.

This study obtains the Gi statistic to analyze the spatial clustering of COVID-19 cases each week independently from April 28 to 26 October 2021. The Gi statistic is calculated using the following equation:(3)$$G_{i}^{*}=\frac{\sum_{j=1}^{n}w_{ij}x_{j}-\bar{X\sum_{j=1}^{n}w_{ij}}}{s\;\sqrt{\frac{\left[n\sum_{j=1}^{n}w^{2}{}_{ij}{\left(\sum_{j=1}^{n}w_{ij}\right)}^{2}\right]}{n-1}}}$$
where N is the incidence, $x_{i}$ and $x_{j}$ are its value in various locations, X is the mean, and $w_{ij}$ is the weight applied to the comparison between locations i and j. The distance-based weight matrix is in accordance with the inverse distance between locations i and j, which is 1/dij.

#### 2.3.3. GWR (Geographically Weighted Regression) Model

Geographically weighted regression (GWR) was used to analyze the relationship between COVID-19 confirmed cases, recovered patients, and suspected cases. The different epidemic situations and medical resources most likely affect the development of a disease in a particular area. Therefore, Brunsdon et al. [31] propose a GWR model, which is an extension of the usual linear regression model by including geographic location data into the regression parameter, as shown in the following equation:(4)$$y_{i}=\beta_{i0}+\sum_{i=0}^{p}\beta_{ik}x_{ik}+\varepsilon_{i}\;$$
where $y_{i}$ is the i-th dependent variable, $x_{ik}$ is the k-th independent variable of location i, p is the total number, and i0 is the intercept parameter at location I, which is the regression coefficient for the k-th at location and varies with the geographic location; i is the error value at location i. The spatial weight matrix in this study is calculated using the bi-square kernel function, as shown in the following equation:(5)$$w_{ij}=({\left(1-{\left(\frac{d_{ij}}{b}\right)}^{2}\right)}^{2}$$

Furthermore, supposing $d_{ij}$ < b, or $w_{ij}$ = 0, where b is the bandwidth, the non-negative attenuation parameter and dij indicates the distance between i-th and j-th observation points. The bandwidth was calculated by optimizing the cross-validation’s square root mean prediction error [32].

#### 2.3.4. Ordinary Least Squares (OLS)

Other indicators of the occurrence of COVID-19 besides confirmed cases, recovered patients, and suspected cases are air pollutants, such as O_3_, CO, SO_2_, and NO_2_, which had spatial homogeneity data. Unlike GWR, which focuses on tackling spatial heterogeneity and does not explicitly address spatial homogeneity, the ordinary least squares (OLS) is often used when the dependent and independent variables do not vary or have spatial homogeneity [33]. Therefore, the OLS method was used to determine the relationship between confirmed cases of COVID-19 and air pollutants.

According to Hutcheson [25], ordinary least regression (OLS) is a generalized linear model performed in ArcGIS software and used to model the selected variables. This is the main technique for analyzing data and serves as the basis for several other approaches, such as the generalized linear model and ANOVA. Therefore, OLS regression is applied in various fields to the exact code of categorical variables and single or multiple variables annotations. At the basic level, the relationship between the response (y) and explanatory variables (x) is represented using the best-fit line, which is predicted, to some extent, by x. In assessing the model, 6 indicators supported the study feedback needed to determine whether the analysis was successful, namely model performance, significance, bias, explanation of the variables, Koenker statistics (BP), and spatial autocorrelation [34].

### 2.4. Air Pollutant Concentration Due to COVID-19

Despite the severe impacts of lockdown on social activities, global mobility, and the economy, there are reports that it temporarily improved environmental conditions. For example, it enhanced the air quality of 103 states in India, the most polluted country globally, with 21 out of the 30 world’s most polluted cities [35]. Furthermore, air pollution is the largest environmental health risk, with 7 million premature deaths yearly. Moreover, more than 91% of the world’s population resides in places where air quality exceeds the World Health Organization’s guidelines [36].

This also causes adverse impacts on the society, economy, and the environment, including climate change. It is, in fact, the major public health, environmental, and developmental challenges of the present generation [37]. Therefore, understanding this temporary improvement in the air quality at the planetary scale provides a unique opportunity to study processes and implications of policy changes in the future. Media reports, primarily based on satellite images, concern a decrease in air pollution due to the global lockdown and some recent scientific studies in a few countries [15,16].

The data from the TROPOspheric monitoring instrument (TROPOMI) sensor on the Sentinel-5 Precursor (Sentinel-5P) was collected and processed in the Google Earth Engine (Google, Mountain View, CA, USA) [38]. The European Space Agency (ESA) generated the Sentinel-5P satellite mission to bridge the data gap between the decommissioned ozone monitoring instrument (OMI) and the scanning imaging absorption spectrometer for atmospheric cartography (SCIAMACHY) onboard satellites Aura and ENVISAT, respectively, and the Sentinel-5 mission’s expected launch. The primary goal of this mission is to prepare the space observations for operational monitoring of air quality, ozone, and surface UV radiation, as well as the climate, by presenting timely atmospheric composition measurements [39]. The TROPOMI sensor onboard the Sentinel-5P satellite is a high-resolution instrument that provides daily worldwide coverage from ultraviolet (UV) to shortwave infrared (SWIR) at selected spectral areas. This enables the recovery of important atmospheric elements, such as (NO_2_), (O_3_), formaldehyde (CH_2_O), (SO_2_), methane (CH_4_), carbon monoxide (CO), aerosol, and clouds. Sentinel-5P has a good signal-to-noise ratio, thus it can be functioned in low-light environments [40].

## 3. Results

### 3.1. Hotspot Clustering

COVID-19 cauterization is defined as an attribute value in each village, with red dots used to denote locations hotspots. The figures estimated by the Gi* statistic show the virus prevalence based on the weekly data from 28 April to 26 October 2021. The mean centers of the virus are weighted based on the incidence per Kelurahan for 26 weeks and marked with green dots. The DD of COVID-19 distribution in the study area was obtained from 30 June to 24 August 2021, marked by the black ellipse, and calculated using the CGD method described in Section 2.3.1.

In Figure 3, the outcome is shown weekly from 30 June to 24 August 2021. COVID-19 is at its peak during this 8-week timeframe (30 June to 24 August 2021). This finding suggests that infection rates fluctuate and that risk patterns may shift over time. The eastern and southern Surabaya regions with high infection rates were found based on the 8 weeks of the COVID-19 peak, 30 June to 24 August (Z scores ranged from 1.508 to 7.172 with a confidence interval of 99%). Meanwhile, North Surabaya has been recognized as a low-infection zone (z-scores ranged from −4.927 to −1.318 with a confidence interval of 99%).

Some urban villages in the north of Surabaya, such as Gunung Anyar, Bulak, and Kembangan Utara, had fewer infections than others from 30 June to 6 July 2021. However, these areas were identified as high-risk from 7 to 13 July 2021, with z score higher than 4.06 at a hotspot of 99% confidence. Another example is the urban villages of Sawahan, Balas Klumprik, Tanah Kali Kewall, Sukolilo Baru, Gading, Jagir, and Kejawan Putih Tambak, identified as insignificant areas from 7 to 13 July 2021, which changed to high-risk areas (hotspot—99% confidence) from 14 to 20 July 2021, as shown in Figure 3. This approach suggests that epidemiologists are able to understand clusters of disease cases assuming the spatiotemporal characteristics used are considered.

### 3.2. Weighted Mean Center (WMC)

Figure 3 used the small green circle to illustrate the weekly changes of COVID-19 incidence in the weighted mean center (WMC) using ArcGIS software, whose purpose has been stated in Section 2.3.1. Within 26 weeks, the x and y coordinates of the average COVID-19 center in Surabaya moved many times. Overall, the results of tracking changes revealed that the center of the COVID-19 outbreak moved randomly over 26 weeks with a confidence interval < 90% at a z-score of 1.386 and a *p*-value of 0.165. The COVID-19 average center was initially located at 691203.3693; 9195225.8514 m; however, a change in the position of WMC occurred over time. For example, on 14 July 2021, WMC was located at 696771.9754; 9192081.5861 m, and a week later, it shifted to 692970.8866; 9193668.8914 m, approximately 3 km to the northwest on July 21, 2021, as shown in Table 2. Then in Figure 3, from August 4 to 18 (6th–8th week), the trend spread moved towards the northeast. This can be further consideration for policymaking, such as the enforcement of restrictions on community activities in the northeastern part of Surabaya.

### 3.3. Directional Distribution (DD)

Directional Distribution (DD) analysis demonstrated that the trend of COVID-19 cases had shifted randomly from various directions in 26 weeks. Figure 3 and Appendix A indicated that in 26 weeks, the Ketintang village and its surroundings were vulnerable to the pandemic. Although the Ketintang village is not significant, this area is a distribution center for three times in 26 weeks (as shown in Appendix A) and a hotspot location with a 99% confidence interval. The ellipse size indicates that the area is vulnerable to virus spread; therefore, it can help in policymaking. The smaller ellipse size at the end of the research period, which is on 26 October 2021, could be influenced by the policies made by the government, such as the imposition of restrictions on community activities, which came into effect again on 22 June 2021, and public order complies with government policies. The government’s policy to impose restrictions on community activities reduced the spread of the COVID-19 delta variant.

The ellipse shape that changed size and shifted during the study period demonstrates the length and width of each ellipse’s axis, rotation, and area, as indicated in Table 2. In 26 weeks, namely from 28 April to 26 October 2021, the orientation and direction of the virus spread showed a spatio-temporal trend, which affected the population. Therefore, the standard deviation of the average center of COVID-19 spread can be determined. For example, the ellipse area was 131 km^2^, 11 km wide, and 14 km long on 30 June 2021, while on 7 July 2021, it changed into 10 km wide, 11 km long, and 92 km^2^, as shown in Figure 3 and Table 2.

### 3.4. Spatial Clustering

Moran’s I (z-score) and General Ord G (z-score) autocorrelation are marked in blue, while Moran’s *p*-value and General G *p*-value are in orange, as shown in Figure 4. Morans I and General Ord G are calculated using spatial integrated statistics discussed in Section 2.3.2.

The *p*-value and z-score have an inverse relationship where the *p*-value is close to 0, with a high z-score indicating a cluster pattern. Moran’s I global statistics (Figure 4a) indicate that the COVID-19 cases in the dataset have a random pattern and are clustered during the spike in cases from July 7th to August 24th, 2021. The *p*-value at the peak of cases, for example between 7 July to 24 August 2021, ranged from 0–0.001, while the z-score value ranged from 3.111–7.867. This demonstrates that the Moran’s I spatial autocorrelation is statistically significant with a confidence interval of 99.9%. After the decline in COVID-19 cases in September, the distribution pattern of cases became random with a *p*-value of 0.139–0.909 and a z-score of −1.235–1.478. The more negative z-score value indicates a random pattern.

General G statistics (Figure 4b) demonstrate the significant height of the clustered COVID-19 incidence. The COVID-19 cases in the dataset were high-clustered several times before the strict social restrictions occurred on 22 July. At the peak of the incidence from 4 August to 14 September, the virus was high clustered with a p-value of 0–0.074 and a z-score of 0.178 to 4.717, at a confidence interval of 99.9%. After the peak of COVID-19 incidence from 15 September to 26 October, the cases decreased, indicating a random pattern of *p*-value—0.142 to 0.673 and z-score—−1.46 to 0.421 with a confidence interval < 90%. This study demonstrates the significant spatial autocorrelation between villages is positively, significantly, and spatially related distance from 7 July to 24 August 2021, as shown in Figure 4. The pattern of COVID-19 became clustered when cases spiked from 7 July to 24 August 2021, indicating that the disease had spread less rapidly. Meanwhile, the delta variant infection decreased in Surabaya from September to October.

### 3.5. Geographically Weighted Regression (GWR) 

Three parameters were used to understand the coefficient spatial distribution of independent variables in the GWR model, discussed in Section 2.3.3 and its correlation from 28 April to 26 October 2021. The parameters are population density, number of recovered patients, and suspected in the previous week. The GWR model was studied to evaluate the heterogeneity of its coefficients in space and predict the spread pattern of COVID-19 one-week ahead shown in Table 3.

Three parameters, consisting of population density, recovered patients, and suspected cases, were used by GWR, of which population density had a negative coefficient value. The recovered patient had a positive correlation, which led to an increase in coefficient at the end of the study period. Suspected people had a negative correlation initially, which became positive in the end. Suspected cases are the most influenced variable for COVID−19 incidence, shown by the largest values between other variables. R^2^ was used to understand the spatial relationship among these factors, which was weak and strong with correlation values of 34% and 96% in the three parameters.

The spatial relationship in week 12, the virus peak, demonstrated that three parameters had a modest correlation (0.55). This model explained about 55% variation in the independent variables responsible for its incidence. The higher coefficient values of independent variables predict a good explanation for the dependent. In suspected people parameters, which are the most influenced factor, several affected zones were found in the northeast, northwest, and southwest parts, as shown in Figure 5c. In population density, strong influence zones were found in the north part of the city (Figure 5a). Meanwhile, in recovered patient parameters, strong influence zones were found in the northeast, east and south parts, as shown in Figure 5b.

### 3.6. Air Pollution Levels Due to COVID-19 Lockdown

This section aims to assess the air pollutant level of CO, NO_2_, O_3_, and SO_2_ in the troposphere by employing remote-sensing data. Sentinel-5P image processing performed with the Google Earth Engine (GEE) resulted in maximum, minimum, and average pollution concentrations in the troposphere during the 26-week study period, as shown in Figure 6.

Tropospheric pollutant concentrations for 26 weeks, 6 weeks before and 7 weeks after the COVID-19 lockdown, were also mapped to observe the temporal variation through remote-sensing data, as shown in Figure 7. All parameters, such as CO, NO_2_, O_3_, and SO_2_ have the most significant impact on pollution concentration when there is a lockdown policy. O_3_ has the highest concentration of 0.11868 mol/m^2^ versus the other air parameters detected throughout the entire region. However, the concentration decreased due to the lockdown period (June 30) policy and increased after 7 weeks of lockdown implementation (started from August 18). Figure 8 shows a graph of each pollutant parameter. CO, O_3_, and NO have the same trend. Before the lockdown until the lockdown gave a decrease trend, while SO_2_ gave a decreasing trend, but not too significant. The trend in all pollutant parameters increased when lockdown conditions were lifted, which was 7 weeks after the implementation of the lockdown (starting from August 18).

The highest SO_2_ concentration of 5.36 × 10^-4^ mol/m^2^ was recorded from September 15th to 21th and indicated no spike in COVID−19 cases. Since the lockdown conditions were lifted (back to the third level from the fourth level) in early September, it was found that the SO_2_ has increased considerably 7 weeks after the lockdown was implemented, which is presented in Figure 7. At the fourth level, transportation capacity was 50%, while at the third level it was 70%. Meanwhile, the industrial sector at PPKM level 3 may operate with a maximum capacity shift setting of 50% for each shift, while the industrial sector at the fourth level can function with a maximum capacity shift setting of 50% for each shift [41].

The increase was significantly observed in the southern part of Surabaya and the city center, which is the main economic activities area. The southern part of Surabaya is also the main entry point for transportation of other city residents, shown in Appendix A. Furthermore, the existing lockdown rules minimized human movement, which went back to normal after the rules were removed. Therefore, industries and transportation that use coal and motor vehicle fuel increased the sulfur dioxide concentration [42].

The highest O_3_ concentration occurred in the final week i.e., from October 20th to 26th, 2021, caused by the spread of pollution from industrial activities. The lockdown has been lifted and industries have resumed its operations in this environment (high O_3_ period). This is in accordance with research by [43], which indicate that the increase in ozone is caused by pollution from the industry; Figure 7 demonstrates that two months after the lockdown, the northern and southeastern research areas dominated by the industry had high O_3_ levels with the highest CO concentration of 3.15 × 10^−2^ mol/m_2_ in the final week. Almost all areas within the center distribution of COVID−19 were contaminated by CO pollution, which led to the major cause of the distribution center.

Furthermore, an increase in air pollutants, such as NO_2_, CO, SO_2_, and O_3_, was observed after 7 weeks of implementation (started from 18 August), despite a decrease in verified COVID−19 cases. Because of the decline of COVID-19 instances, the government decided to relax the lockdown policy guidelines on 24 August. This demonstrates that the presence of COVID−19 cases, the lockdown policy, and air pollution are all linked.

### 3.7. Relationship between COVID−19 Incidence and Air Pollutant

The Kriging-interpolation for Sentinel-5P imagery processed using the Google Earth Engine (GEE) was used to conduct this research. The OLS regression results for all parameters are shown in Table 4. It was positive with a statistically significant correlation between COVID-19 cases and air pollution during the study period. The increasing value of r between 14 to 28 July depicts results in accordance with the incidence data in Figure 2, where the largest r value is 0.3331. This is in accordance with the state of the highest number of COVID-19 cases in that week. These indicate that there may be a connection between positive cases and fine air pollution (O_3_, CO, NO_2_, and SO_2_) with the greatest effect by NO_2_. This relationship is indicated by the large coefficient value of −1199337.944, obtained in 12th week on 14 July.

## 4. Discussion

This study was conducted using different spatiotemporal and statistical methods, such as CGD analysis, patterns, clustering, and relationships between variables, which are relevant in understanding COVID-19 spread in Surabaya from 28 April to 26 October 2021. The first approach applied involved analyzing the geographic distribution of the virus. The weighted mean center changed throughout the study phase and was denoted with a small circle highlighted in green, as shown in Figure 3. The ability to ascertain its weight is critical for tracking the value of variations associated with an area. The results obtained revealed dynamic regions that change over time due to the spread of the virus to several communities in Surabaya. According to [13], several factors can transmit viruses, including air pollution, geo-meteorological and social parameters. The high number of cases associated with its spread in Indonesia, especially in East Java, prompted the government to impose a lockdown policy. Most Indonesians infected with the virus possess the delta variant [44].

The DD trend is used to obtain the COVID-19 ellipse coincides with the affected area’s population size and density features. Besides, these are partly found in the eastern and southern parts due to various government activities, residents, and schools located in these regions. Meanwhile, the northern part is a cold spot not included in the elliptical, swamp, and coastal areas, which are not dense. All these circumstances have proven that the virus spread is based on identifying its spatiotemporal pattern, which is higher in the most densely populated areas [12].

The Moran’s I result also demonstrated a positive significant spatial autocorrelation of the virus. The spike in COVID-19 incidence affected the number of positive cases in several areas in Surabaya, as evidenced by the results of Moran’s I *p*-value, which is closer to 0. Spatial data are simply described as highly correlated, assuming likely values are spatially close to each other [45]. However, the Moran’s I autocorrelation did not identify whether the distribution pattern was high or low. Therefore, this study applied the G test and demonstrated a significant positive correlation to validate and identify strong spatiotemporal patterns in the association between the factor and COVID-19 infection. Other factors, such as confirmed cases, the number of recovered patients a week prior, and the number of suspected people in the previous week, were also used to identify the COVID-19 pattern.

The relationship between air pollution concentrations and the number of COVID-19 cases using OLS did not demonstrate a significant pattern, as indicated in Table 4. At the end of the study period, the number of cases decreased by the ellipse area due to the narrowing level of air pollution distribution. The area in the ellipse, which is the distribution center of most COVID-19 cases, has a low level of pollution and mobilization. This is because the implementation of lockdown rules was tightened in areas with a high case rate. Areas without strict social restriction rules experience high vehicle mobilization and air pollution levels. According to Qin et al. [46], people living in regions with poor air quality are highly vulnerable to COVID-19 due to the long-term inhalation of toxic pollutants.

## 5. Conclusions

The COVID-19 data obtained in Surabaya from 28 April to 26 October 2021 used the spatial CGD, Moran’s I, General-G, Gi* statistics, and the GWR models to conclude that the delta variant has a significant spatial correlation with the variables. Although Moran’s global I and G statistics were used to identify strong spatial patterns of the virus regarding the variables, this approach only considers single-layer distributions at any given time. This study was able to identify which village demonstrated a high probability of infection using Gi* hotspot and cold spot analysis. Therefore, future studies need to investigate other correlations, such as ecological, climatological, and socioeconomic variables, to effectively determine the relationship between COVID-19 hotspots, cold spots, and population density. Although the transmission is currently showing a declining trend, the epidemic situation in eastern and southern Surabaya is difficult. The spatiotemporal analysis demonstrated in this study suggests that a temporal hazard model based on weekly infection rates led to a better understanding of changes. The pollutant processing results with the number of confirmed cases of COVID-19, especially the delta variant, demonstrated a fairly strong positive correlation value of 0.577 in the third week, which was the highest. Therefore, COVID-19 is correlated with pollution levels. This study is expected to provide a useful strategy in improving the infectious disease surveillance system and control intervention in each affected area for future purposes. GIS was used to map disease incidence against several parameters, including demographics, environment, geography, and past events, to understand outbreak origins, spread patterns, and intensity, supporting the implementation of control, preventive, and surveillance measures.

There are several caveats and inherent limitations in this study. First, Sentinel-5P has a low geographical resolution compared to village areas, demanding the usage of medium resolution satellite photography. Sentinel-5, on the other hand, has a unique mission and sensor for monitoring the troposphere on a city-scale, allowing the researcher to collect data on O_3_, CO, SO_2_, and NO_2_. Due to the highly dynamic nature of the disease, such as the highly contagious delta variant, it is necessary to add an analysis of the vaccination rate in each region to reduce the number of distributions and positive cases. For future studies, adding vaccination variables for each region can be added to see the rate of infection and the spread of COVID 19.

## Figures and Tables

**Figure 1 ijerph-19-01614-f001:**
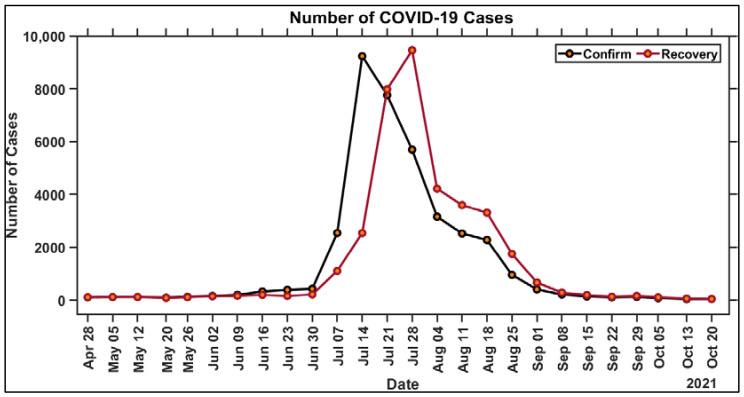
Graph of COVID-19 cases for 26 weeks in Surabaya.

**Figure 2 ijerph-19-01614-f002:**
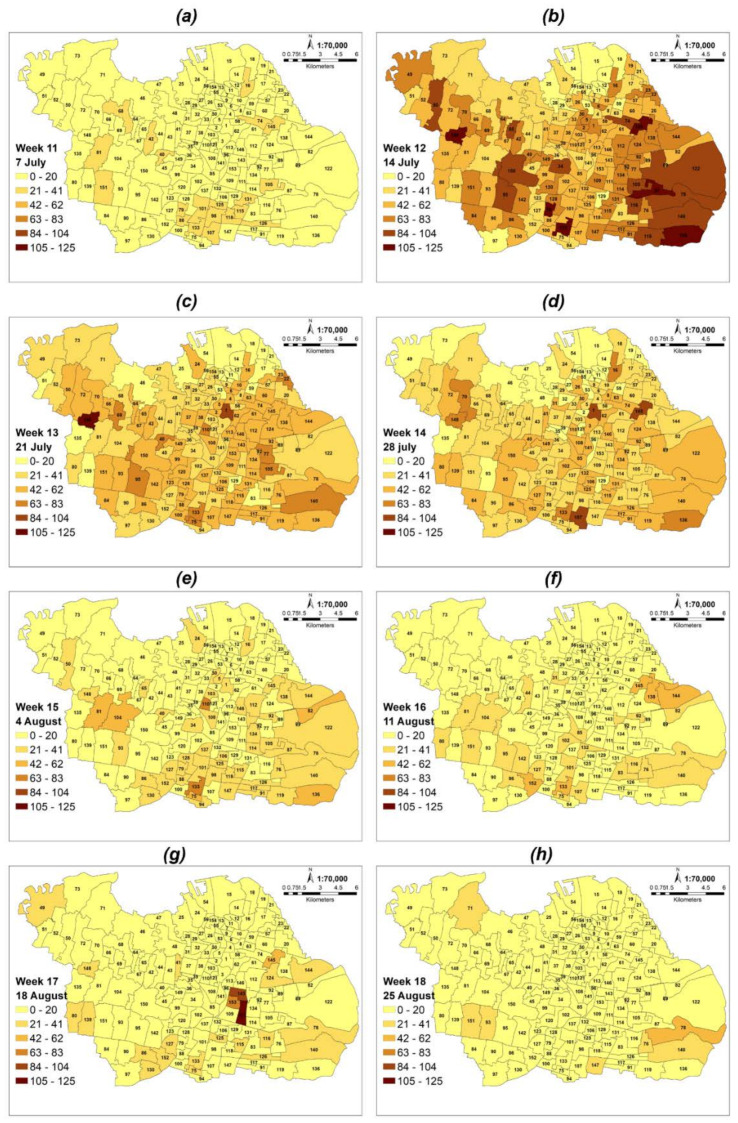
The peak distribution of confirmed COVID-19 cases per 100,000 populations every week in Surabaya (complete period figure for 26 weeks can be accessed in Appendix A. (**a**) 11th Week (7–13 July).; (**b**) 12th Week (14–20 July.; (**c**) 13th Week (21–27 July).; (**d**) 14th Week (28 July–3 August).; (**e**) 15th Week (4–10 August).; (**f**) 16th Week (11–18 August).; (**g**) 17th Week (19–24 August).; (**h**) 18th Week (25–31 August).

**Figure 3 ijerph-19-01614-f003:**
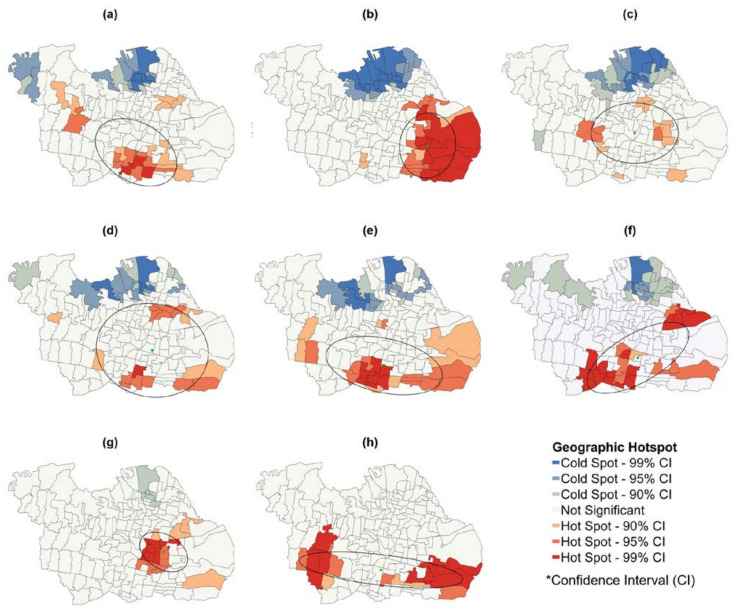
The peak clustering map of COVID-19 incidence along with its orientation and shift directions (complete period figure for 26 weeks can be accessed in Appendix A). (**a**) 11th Week (7–13 July).; (**b**) 12th Week (14–20 July).; (**c**) 13th Week (21–27 July).; (**d**) 14th Week (28 July–3 August).; (**e**) 15th Week (4–10 August).; (**f**) 16th Week (11–18 August).; (**g**) 17th Week (19–24 August).; (**h**) 18th Week (25–31 August).

**Figure 4 ijerph-19-01614-f004:**
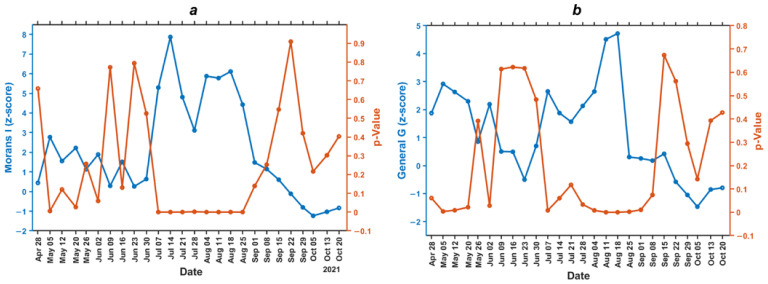
Spatial autocorrelation graph. (**a**) *p*-value and z-score Morans’s I correlation; (**b**) *p*-value and z-score General G correlation.

**Figure 5 ijerph-19-01614-f005:**
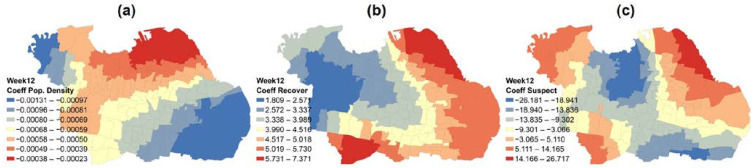
Spatial distribution of influence in (**a**) population density, (**b**) recovered patient, and (**c**) suspected cases in the peak of COVID-19.

**Figure 6 ijerph-19-01614-f006:**
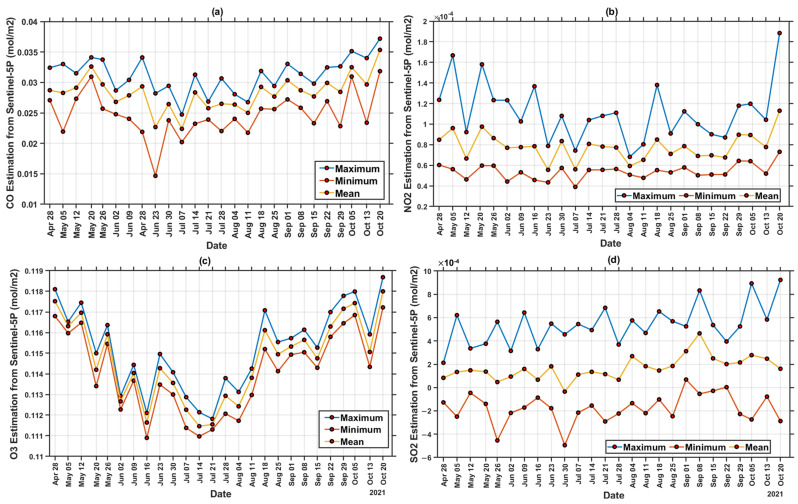
Concentration values of (**a**) Carbon Monoxide (CO); (**b**) Nitrogen Dioxide (NO_2_); (**c**) Ozone (O_3_); (**d**) Sulfur Dioxide (SO_2_).

**Figure 7 ijerph-19-01614-f007:**
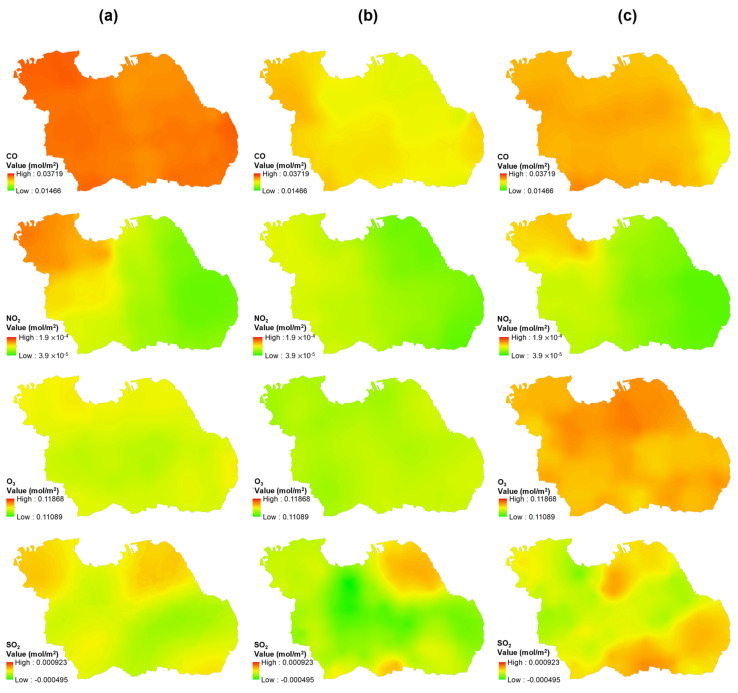
Air parameter concentration (CO, NO_2_, O_3_, and SO_2_) in Surabaya: (**a**) a month before, (**b**) next day, and (**c**) a month after region lockdown.

**Figure 8 ijerph-19-01614-f008:**
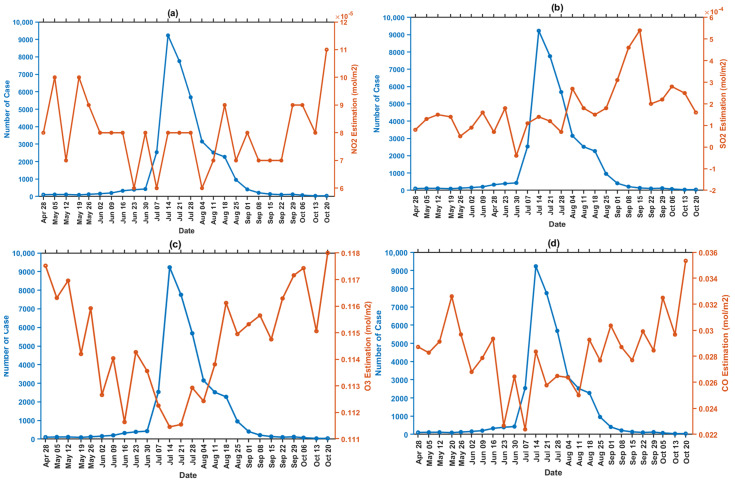
Graph correlation of COVID-19 incidence and air parameter based on Sentinel-5P image processing in GEE per week. (**a**) NO_2_ tropospheric concentration, (**b**) SO_2_ tropospheric concentration, (**c**) CO tropospheric concentration, and (**d**) O_3_ tropospheric concentration.

**Table 1 ijerph-19-01614-t001:** The dataset was obtained from BPS, BIG, OSM, and Satuan Tugas (task force) COVID-19 Surabaya.

No	Data	Source
1	Population	Badan Pusat Statistik Surabaya 2020
2	Population density	Badan Pusat Statistik Surabaya 2020
3	Surabaya administrative boundaries (city and village)	Badan Informasi Geospasial dan Open Street Map
4	COVID-19 daily data (confirmed, recovered, suspect)	https://lawancovid-19.surabaya.go.id/ (30 June 2021)
5	Air Pollution (NO_2_, SO_2_, O_3_, and CO)	Google Earth Engine (Sentinel-5P)

**Table 2 ijerph-19-01614-t002:** Changes in DD from center-weighted mean confirmed cases per 100,000 populations (incidence) COVID-19 over a 26 weeks study period.

Week	Date	Length (km)	Width (km)	Area (km^2^)	Rotation
1	28 April 2021–4 May 2021	4.94367	1.75906	27.32002	64.34984
2	5 May 2021–11 May 2021	5.96700	2.43495	45.64533	104.39636
3	12 May 2021–19 May 2021	6.10525	3.35143	64.28118	95.09706
4	20 May 2021–25 May 2021	5.32776	4.12860	69.10306	96.81598
5	26 May 2021–1 June 2021	7.19094	2.99919	67.75481	73.92413
6	2 June 2021–8 June 2021	8.20458	2.37165	61.13025	104.63646
7	9 June 2021–15 June 2021	6.10076	1.34384	25.75626	49.12235
8	16 June 2021–22 June 2021	10.37618	4.00459	130.54036	123.65276
9	23 June 2021–29 June 2021	3.62656	0.73228	8.34304	62.94312
10	30 June 2021–6 July 2021	5.08497	6.69721	106.98736	42.41460
11	7 July 2021–13 July 2021	5.47192	3.65635	62.85471	120.54456
12	14 July 2021–20 July 2021	3.48135	4.07568	44.57561	9.36094
13	21 July 2021–27 July 2021	5.42577	3.75721	64.04384	90.09924
14	28 July 2021–3 August 2021	7.00457	5.80578	127.75903	100.71435
15	4 August 2021–10 August 2021	7.38358	3.36916	78.15178	102.26742
16	11 August 2021–17 August 2021	7.12187	3.01337	67.42110	59.90758
17	18 August 2021–24 August 2021	3.24594	2.13313	21.75242	120.95107
18	25 August 2021–31 August 2021	10.26110	1.89992	61.24630	96.76604
19	1 September 2021–7 September 2021	7.91130	1.57939	39.25427	93.08196
20	8 September 2021–14 September 2021	7.65780	1.92917	46.41125	78.09290
21	15 September 2021–21 September 2021	5.44430	1.21007	20.69670	48.95904
22	22 September 2021–28 September 2021	1.98095	9.99348	62.19288	139.57942
23	29 September 2021–5 October 2021	1.88549	6.52740	38.66456	33.70470
24	4 October 2021–12 October 2021	0.98472	1.98604	6.14400	143.11851
25	13 October 2021–19 October 2021	1.87011	4.72545	27.76268	177.96124
26	20 October 2021–26 October 2021	0.96843	2.19412	6.67537	0.80506

**Table 3 ijerph-19-01614-t003:** Model-fitting results derived from the geographical weighted regression (GWR) demonstrating the relationship between COVID- 19 incidence (dependent variable) and 3 independent variables, namely population density, recovery patient, and suspected cases.

Week	Estimated Coefficient			Standard Error			R^2^
	Pop_Density	Recovery	Suspect	Pop_Density	Recovery	Suspect	
1	−0.000006	−0.00145	0.5942	0.000004	0.00081	0.0827	0.34
2	−0.000006	1.58807	−0.3376	0.000005	0.44246	0.6092	0.08
3	−0.000006	1.44143	0.7252	0.000007	0.50798	1.1458	0.37
4	−0.000003	0.9533	−0.1323	0.000005	0.46152	0.7389	0.14
5	−0.000012	1.76234	0.435	0.000005	0.50614	0.8879	0.11
6	−0.000008	1.62216	−0.1898	0.000009	0.61543	0.3095	0.32
7	−0.000001	0.23091	−0.2002	0.000007	0.52738	0.174	0.01
8	−0.000008	0.35447	−0.0628	0.000009	0.48646	0.1758	0.01
9	0.00001	−0.00515	0.1042	0.000011	0.77978	0.1664	0.05
10	0.000008	0.07483	−0.3727	0.000014	0.89865	0.3131	0.01
11	−0.000038	3.56064	−2.4404	0.000045	0.75624	3.2563	0.41
12	−0.000644	4.32694	−5.6831	0.000179	0.96196	17.562	0.55
13	−0.000467	1.13803	−1.3819	0.000115	0.30656	2.2285	0.30
14	−0.000347	1.40317	−1.176	0.000143	0.33388	1.2351	0.46
15	−0.000306	0.84206	−6.6564	0.000133	0.38841	13.029	0.43
16	−0.000228	1.87945	−5.2903	0.00011	0.37022	12.44	0.72
17	−0.000129	3.09936	10.063	0.00015	0.62592	18.485	0.68
18	0.00005	3.03727	3.4316	0.000038	0.3922	0.4289	0.89
19	−0.000056	3.83066	4.0997	0.00003	0.60965	0.7658	0.86
20	−0.000035	3.80442	4.3494	0.000015	0.51624	0.6211	0.85
21	−0.000007	4.4022	4.9018	0.000011	0.54769	0.6049	0.88
22	−0.000026	5.63802	5.2415	0.000014	0.73245	0.7864	0.94
23	−0.000014	4.5608	5.9564	0.000008	0.49729	0.6306	0.79
24	−0.000008	4.43153	4.6766	0.000003	0.38015	0.4695	0.76
25	−0.000004	4.37876	4.3851	0.000003	0.42768	0.5624	0.72
26	−0.000006	6.40907	7.2381	0.000004	0.38539	0.7607	0.96

**Table 4 ijerph-19-01614-t004:** OLS regression between COVID−19 incidence and air pollution parameters.

Week	r	R	Coefficient			
			O_3_	CO	SO_2_	NO_2_
1	0.059	0.243	41.078	−115.823	2196.099	−8845.032
2	0.035	0.188	183.530	−9.763	−908.882	−3114.051
3	0.006	0.077	−294.505	21.238	−455.957	−976.079
4	0.069	0.262	114.451	163.894	−774.772	−7748.330
5	0.074	0.272	−148.850	155.290	191.222	−16,298.614
6	0.084	0.289	1564.527	−237.265	1601.451	4634.166
7	0.015	0.123	33.325	119.347	−200.555	−2408.472
8	0.023	0.151	−381.530	−20.501	843.237	7701.886
9	0.011	0.107	−511.875	26.044	−420.067	7561.081
10	0.019	0.137	1464.636	219.159	795.614	2537.760
11	0.103	0.322	4849.546	1036.376	6103.810	−223,496.785
12	0.333	0.577	−31,165.367	4160.563	88,138.945	−1,199,337.944
13	0.123	0.350	27,404.369	−13,248.590	−6139.623	−13,213.022
14	0.070	0.265	2107.077	5268.257	23,460.160	−614,879.134
15	0.235	0.485	6979.684	−4141.370	−43,595.925	−328,846.454
16	0.008	0.092	1472.020	3265.895	3536.649	−336,158.858
17	0.117	0.341	−13,664.796	−481.837	5318.525	−106,384.125
18	0.054	0.233	−4594.035	424.550	4146.287	−24,345.855
19	0.054	0.233	−2347.285	276.720	−462.544	−24,110.595
20	0.021	0.146	452.573	−44.539	−1317.076	14,171.958
21	0.064	0.253	396.631	−310.415	370.013	−15,564.079
22	0.047	0.216	971.238	−222.039	−2481.582	36,013.294
23	0.028	0.166	−741.397	−27.302	1313.003	556.694
24	0.082	0.286	442.884	95.709	−204.715	−4617.635
25	0.040	0.199	−335.258	23.522	−760.015	−636.684
26	0.027	0.166	159.888	88.328	−593.480	3458.184

## Data Availability

The Surabaya population and density data can be accessed in Badan Pusat Statistik (BPS) Surabaya 2020. The Surabaya administrative boundaries was accessed in the Badan Informasi Geospasial dan Open Street Map. The COVID-19 daily data by Satuan Tugas COVID-19 Surabaya was accessed in https://lawancovid-19.surabaya.go.id/ (accessed on 30 June 2021).

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
