# Peer review of "Spatiotemporal Analysis for COVID-19 Delta Variant Using GIS-Based Air Parameter and Spatial Modeling"

_ijerph, 2022, doi:10.3390/ijerph19031614_

Round 1
Reviewer 1 Report
The manuscript investigated the spatiotemporal pattern of the spread of Covid-19 in Surabaya. It shows the correlation between Covid-19 confirmed case number and population density, number of recovered patients, and number of suspected people. Moreover, this paper also shows the correlation between the number of Covid-19 cases and the concentration of air pollutions. The results of this study can improve our knowledge of the Covid-19 spread pattern, which can benefit governors to develop better Covid-19 policies and guidance. Overall, this manuscript is not well written. Many cases require additional information to clarify motivation, methodology, results, and interpretation. Furthermore, there are many grammar and spelling problems, making the paper very difficult to read. It requires professional proofreading before it is resubmitted for publication. Please note that my review is not affected by the language, and my entire focuses are on science. My impression of this paper is that it could be improved by considering the following suggestions. The revised paper should discuss these points, not just answer in the authors' response. I am willing to review the revised manuscript. Therefore, my recommendation for the editor is that this manuscript needs major revision.
General comments:
- Figures are blurry. Please use a higher resolution.
- It is not a good idea to use EXCEL to make plots for publication. Please consider using other software such as Python, R, or OriginaLab.
- Please add the necessary references. There are lots of places that miss references. For example, L53-55, “COVID-19 is … disease spread.” Please check the entire manuscript carefully and add missed references. Otherwise, this manuscript can be considered plagiarism.
- Please spell out all abbreviations the first time they appear on the paper.
Specific comments:
- L129-130, “This study … in Surabaya.” You should mention the studied period here. This is not clear until very late in this paper.
- L135-137, “Fig 3 shows … in Fig 4.” First, this does not match the data shown in Fig 3. This is not the only place you made this mistake. Please check the entire manuscript and correct either text or the figure. Second, are Fig 3 and Fig 4 show new cases, accumulated cases, or total positive cases? It is not clear in the paper.
- L138-140, “We also use … of COVID19.” First, what is the data source of non-pharmaceutical interventions? Second, I did not see any discussion of that. These can be very interesting and useful. Please add a discussion about the correlation between non-pharmaceutical interventions and confirmed Covid 19 cases.
- Figures 1 and 2 can be moved to supplementary information.
- Table 1. Please reorganize the table. No is not necessary. Also, what does format mean here? Do you mean the format of these data when you received them? Why is format important?
- Figure 3. I suggest adding the actual date. It is tough for me to figure out the actual time of each Epi week.
- Section 2.3.4. I suggest adding a correlation test between confirmed Covid 19 cases and all investigated variables.
- L262-L265, “Indicators … NO2).” What is the connection between these two sentences?
- Section 2.4. This section should explain the source of air pollution data, not a literature review of air pollution.
- L302-315, “The results presented …as consideration.” Do you have any explanations for why you observed these changes? What can you learn from your result?
- L320-322. “Overall, … 8-week period.” I do not think the center moved randomly. The center is always at the lower part of the figure, and there are more hot spots at the lower part. You should do some statistic analysis rather than look at the plots.
- Figure 5. What does CI stand for?
- L335-338, “As stated in the note, …, 11 August.” What note do you refer to there? Moreover, I am curious why Warugung Village became a hotspot several times.
- Figure 6. It seems your y-axis labels in each plot are inverted. Moreover, what is your x-axis ticks represent? From January 1 to 30, 2021?
- L372-375, “Moran’s Z … 95%).” Where are these data?
- Table 4. Please consider making plots instead of using a table.
- Figure 8. For the same pollutant, please use the same color bar scale. It is not easy to compare concentration and distribution at different events.
- L431-435, “On July 14,…period.” What does this tell you? I have noticed that the average CO concentration during the lockdown week was lower than the lowest CO concentration before lockdown. It seems that locking down will reduce CO emissions. It is also shown in figure 9. Moreover, are you trying to say CO spread entire region will enhance the spread of COVID-19? How?
- L439-440, “The highest … cases.” Figure 9(b) shows SO2 concentration increased significantly after the peak of Covid -19 cases. You should discuss that. Why does that happen?
- L441-442, “Furthermore, … widespread pollution.” Please provide evidence if this is correct.
- L445-447, “In addition, … cases.” It seems all pollutants increased after the case was reduced. Also, during the last week of your study period, NO2 and O3 were increased, not decreased.
- L463-467, “The result … rate [32].” How do you get this result? Also, this seems not fit here.
- L516-526, “Then, the estimated … al. [34].” Why do you mention in situ data here? You are not using it in the paper. If Sentinel-5P data do not match or are not close to in situ data, you need to discuss which one is more accurate, and I will trust in situ date more. Also, how many stations are in Surabaya? If there are more stations, why don't you use all of them?
- L529-531, “In terms of … imagery.” Does that mean there is a higher resolution, but you chose not to use it?
- L531-534, “Last, due … COVID-19.” I do not understand this. Do you mean adding an analysis of vaccination rates can improve your model or reduce the spread of the virus? If it is the first case, why don’t you add that?
Author Response
We deeply appreciate constructive comments by the reviewers. We attach our replies to the individual comments in blue fonts.
Authors

Reviewer 2 Report
Dear Authors:
The aim of paper entitled “Spatiotemporal Analysis for COVID-19 Delta Variant Using 2GIS-based Air Parameter and Spatial Modelling” was very important and relevant resarch problem. After reviewing this article, I have suggestion as follow:
In my opinion, the Study Area should be expanded. I suggest why the city was chosen for the study. there is a characteristic of the city but in my opinion it is not enough. I suggest you add an explanation why the city Surabaya was chosen for the study. there is a characteristic of the city but in my opinion it is not enough.
Reviewer
Author Response

(The authors gave the same response as above.)

Round 2
Reviewer 1 Report
I think the revised manuscript is good for publishing.